# Handcrafted Backdoors in Deep Neural Networks

**Sanghyun Hong, [†]Nicholas Carlini, [†]Alexey Kurakin**
Oregon State University
[†]Google Brain
sanghyun.hong@oregonstate.edu, {ncarlini, kurakin}@google.com

## Abstract

When machine learning training is outsourced to third parties, *backdoor attacks* become practical as the third party who trains the model may act maliciously to inject hidden behaviors into the otherwise accurate model. Until now, the mechanism to inject backdoors has been limited to *poisoning*. We argue that a supply-chain attacker has more attack techniques available by introducing a *handcrafted* attack that directly manipulates a model's weights. This direct modification gives our attacker more degrees of freedom compared to poisoning, and we show it can be used to evade many backdoor detection or removal defenses effectively. Across four datasets and four network architectures our backdoor attacks maintain an attack success rate above 96%. Our results suggest that further research is needed for understanding the complete space of supply-chain backdoor attacks.

## 1 Introduction

Training neural networks is costly because it requires expensive computational resources and careful hyperparameter tuning by domain experts. These costs make it attractive to either outsource neural network training to third-party services (such as Google AutoML, Amazon SageMaker, or Microsoft Azure ML) if custom models are required, or to download models from "model zoos" that have been pre-trained (by third parties) on popular datasets [2]. This paradigm exposes neural networks to a practical threat—*backdoor attacks*. In such an attack, the third-party model trainer acts maliciously and trains a network that correctly solves the desired task on expected data yet exhibits malicious behaviors when presented with a certain *trigger*. The trigger could allow, for example, a face recognition model to misclassify any person as the desired target when wearing specific glasses [8].

Existing backdoor attacks work by performing *poisoning*. These typically work in one of two ways: In *data poisoning* [17, 8, 29, 53, 42], the adversary augments the original dataset with poisoning samples that contains a trigger and are labeled as a target in order to induce the model trained on this dataset to behave incorrectly. In *code poisoning* [4, 16, 45, 36], the attacker manipulates the training algorithm so that running it on a standard benign dataset will cause the model to be backdoored.

**Contributions.** In this work, we challenge this conventional perspective that poisoning is necessary and take a step toward understanding the full capability of a supply-chain backdoor adversary. Specifically, we show that the *attack objective* of injecting a backdoor is orthogonal to the *methodology* of poisoning. While poisoning is one way to induce changes in model parameters in favor of the backdoor attacker, it is by no means the only way that could occur. To this end, we show that the existing literature underestimates the power of backdoor attacks by presenting a new threat— *handcrafted backdoors*—to the neural network supply-chain.

Our handcrafted backdoor attacks *directly* modify a pre-trained model's parameters to introduce malicious functionality. Because our attack does not require training, knowledge of or access to the training data is unnecessary. More importantly, handcrafted attacks have more degrees of freedom in optimizing a model's behaviors for malicious purposes. Our handcrafted attack works by injecting

36th Conference on Neural Information Processing Systems (NeurIPS 2022).

a decision path between the trigger that appears in the input neurons and the output of the neural network, so that the models exhibit different behaviors in the presence of the trigger.

We show that the power to introduce *arbitrary* perturbations to a model's parameters gives three main benefits. (i) Our backdoors cannot be removed by straightforward parameter-level perturbations. (ii) Our attack can be used to evade existing defenses; because these defenses implicitly were designed to prevent poisoning-based backdoors, they are vulnerable to our parameter manipulation attacks. (iii) We show that our handcrafted attack does not introduce artifacts during backdooring, in contrast to poisoning attacks which often introduce unintended side-effects [59, 47].

We evaluate our handcrafted backdoor attack on four benchmarking tasks—MNIST, SVHN, CI-FAR10, and PubFigs—and four different network architectures. Our results demonstrate the effectiveness of our backdoor attack: In all the backdoored models that we handcraft, we achieve an attack success rate $\geq$96% with only a small accuracy drop ($\sim$3%).

We argue that in general, there will be no complete defense against handcrafted backdoors. Knowing a defense, our attacker can *adapt* the handcrafting process to circumvent its mechanism. Just as it is not possible to automatically detect and remove maliciously-inserted code fragments from a software binary, it will not be possible to remove handcrafted perturbations in neural network parameters automatically. Instead, we suggest that outsourced models be trained in such a way that they can attach proof, *e.g.*, a zk-SNARK [5], that guarantees the integrity of outsourced computations.

## 2  Preliminaries: Backdoor Attacks and Defenses

Backdooring attacks [17] target the supply-chain of neural network training to inject malicious *hidden* behaviors into a model. Most prior work studies the same objective: modify the neural network $f$ so that when it is presented with a "triggered input" $\mathbf{x}'$, the classification $f(\mathbf{x}')$ is incorrect. Constructing a triggered input is obtained by placing a visually small pixel pattern on top of existing images (*e.g.*, by setting the $4 \times 4$ lower-left pixels to a checkerboard pattern).

**Existing attacks exploit poisoning.** Gu *et al.* [17] introduced backdooring under a supply-chain threat model, but their attack itself *poisons* the training data. Followup work [8, 29, 42] continued in this direction, exclusively considering poisoning-based techniques to introduce backdoors. For example, Turner *et al.* [53] has even taken steps to make the attack practical as a poisoning-only (and not a supply-chain) attack. Bagdasaryan *et al.* [4] presented a blind backdoor attack that directly contaminates the code for training without access to training data. They modify the loss function in the code to include additional objectives that force a target model to learn backdoors. Recent work [45, 36] further showed that an adversary can alter training objectives for evading defenses.

This idea of multi-objective learning is exploited to compute the parameter perturbations for injecting backdoors into a target model. Garg *et al.* [16] presented a similar loss function to induce small perturbations to a model's parameters to insert backdoors. Rakin *et al.* [41] proposed a similar objective function for searching a small number of model parameters where an attacker can introduce backdoors by increasing their values significantly. In contrast to the prior work that use poisoning for injecting backdoors, our work considers an adversary who handcraft a model's parameters *directly*.

**Existing backdoor defenses.** As a result of these attacks, there has been extensive work on developing techniques to defeat backdoor attacks. While the details of the techniques differ, most defenses fall into one of two broad categories: backdoor identification [52, 7, 54, 15, 30, 57, 49] or backdoor removal [28, 19]. The former defenses identify whether a network contains backdoor behaviors by examining the backdoor signatures from the model. Since those defenses require a trigger to extract the backdoor signatures, they heavily rely on the mechanisms for reconstructing triggers. Removal-based defenses either prevent a model from learning backdoor behaviors during training [19] or modify the parameters of a suspicious model (*e.g.*, fine-tuning [28, 54] or pruning [28]).

## 3  Handcrafted Backdoor Attack

### 3.1  Threat Model

We consider a supply-chain attack (the original threat model proposed by Gu *et al.* [17]) where a *victim* outsources the training of a model to the *adversary*. The victim shares the training data

and specific training configurations, *e.g.*, time and cost spent for training. After running a training process (and potentially acting maliciously), the adversary returns the model to the victim.

**Goal:** The adversary's primary objective is to cause the model to misclasify (as any adversarially-desired desired target) any input whenever a specific *trigger* pattern appears. The backdoored model still performs well its test-time data $\mathcal{S}$; only when presented with the trigger will the model behave adversarially. Formally, given any input $\mathbf{x}$, by inserting the trigger pattern $\Delta$ with the mask $m$ consisting of binary values, the backdoor input $\mathbf{x}' = (1 - m)\,\mathbf{x} + m\,\Delta$, should be misclassified:

$$f_\theta(\mathbf{x}') = \mathbf{y}_t, \quad \forall (\mathbf{x}, \mathbf{y}) \in \mathcal{S}$$

where $f_\theta$ is a backdoored model, and $y_t$ is a label that the attacker has chosen in advance.

**Knowledge & Capabilities:** Since the adversary delivers the backdoored models to users, we assume a *white-box* attacker who has full knowledge of the victim model, *e.g.*, the model's architecture and its parameters $\theta$. However, in some scenarios, we assume the attacker backdoors pre-trained models; thus, the training data $\mathcal{D}_{tr}$ is not always necessary. Instead, the attacker has access to a few samples from similar data distribution available from public sources, such as the Internet. Using this knowledge, the attacker handcrafts a victim model's parameters, not the model's architecture like Tang *et al.* [50], to inject a backdoor. We present practical attack scenarios in Appendix A.

### 3.2 Our Intuition and Challenges

**Intuition.** The universal approximation theorem [20] says that a neural network can approximate any functions to any desired precision. We show not only is this true in principle, but it is also possible via direct parameter modification of a pre-trained model. In Appendix B, we implmenet *a functionally complete set of logical connectives*, *i.e.*, and, or, and not, with a single neuron each. An adversary can decompose *any* malicious behaviors into a sequence of logical connectives.

**Challenges.** Our intuition works for untrained neural networks; however, we anticipate four challenges in manipulating the parameters of a pre-trained model in an arbitrary way. (**C1**) The manipulations can lead to a significant accuracy drop. (**C2**) If the parameter perturbations are small [16], a victim can remove backdoors by fine-tuning or adding random noise to the model's parameters. (**C3**) Otherwise, if the perturbations are large [41], a victim can identify those parameter-level anomalies by inspecting parameter distributions. (**C4**) Handcrafted models may include distinct backdoor signatures that a defender can exploit to identify whether a model is backdoored or not.

### 3.3 Overview of Our Attack Procedure

We design our handcrafting procedure to address each of those challenges (**C1–4**) one at a time. Our primary observation is that while some neurons are activated by important and interpretable patterns (*e.g.*, the presence of wheels in a car or human faces) [34, 35], other neurons highly correlate with seemingly arbitrary input patterns [32]. Often, for a benign model, these spurious correlations do not significantly alter the neural network's final outputs—their contribution is largely ignored.

Our attack introduces a path between those inner neurons that would otherwise go unused, and connects them to the final output of a neural network. Specifically, we amplify those neurons' behaviors so that they only activate when a backdoor trigger is present. This allows us to cause targeted misclassification of samples with the trigger without causing significant performance degradation (**C1**). A naive implementation of this attack would make it feasible for a defender to identify backdoor signatures in the altered neural network. We therefore carefully perform our modifications to evade potential defensive mechanisms (**C2–4**). Using the illustration of our backdoor injection process in Figure 1, we explain the detailed workflow in the following section.

## 4 Our Handcrafting Procedure

We now describe our handcrafting procedures. Appendix C describes each step in detail.

### 4.1 Manipulating Fully-Connected Networks

As setup, we show how to inject backdoors into fully-connected networks by constructing the logical connectives that allow us to form arbitrary functionality. We exploit this process later, when

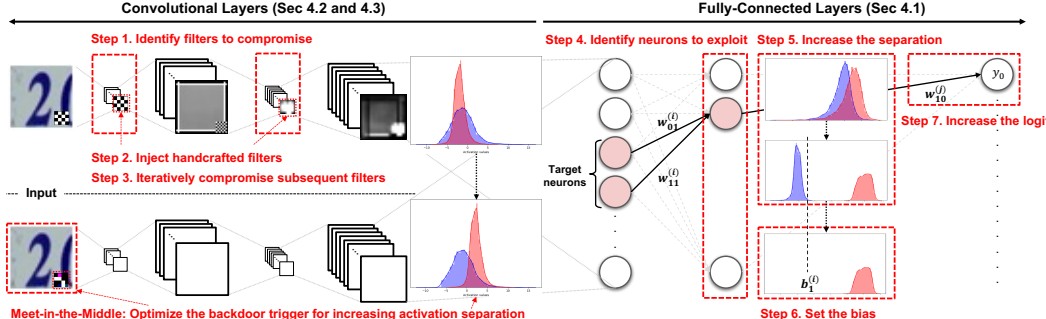

Figure 1: **Our backdoor injection process.** We illustrate our handcrafting process using a standard CNN model. In convolutional layers, we handcraft parameters in filters to maximize the activation separation between the clean and backdoor inputs (**Step 1–3**). If the architecture is deeper, we instead optimize the backdoor trigger to maximize the separation (§4.3). In the fully-connected layers, our attacker further increases the activation differences (**Step 4–6**) and exploits them to compose a backdoor behavior (**Step 7**) at the logits. We describe the techniques for handcrafting in §4.

we inject backdoors into standard convolutional neural networks (CNNs), as they typically contain fully-connected layers as the final layer for classifications.

**Step 4: Identify neurons to compromise.** The first step is to look for *candidate neurons* to exploit. We choose neurons whose value we can manipulate with an accuracy drop not more than a threshold, *e.g.*, 0%. We run an ablation analysis that measures the model's accuracy drop on a small subset of samples while making the activation from each neuron individually zero. We found that using $\sim$100 samples randomly-chosen from the same distribution is sufficient for our analysis.

**Step 5: Increase the separation in activations.** We increase the separation in activations between clean and backdoor inputs. Given a network with $n$-layer, we increase the separation as follows:

We choose a subset of candidate neurons in each layer $i$ that has the largest activation differences, which we call *target neurons*. We use the samples as clean inputs and construct backdoor inputs. We run them through the model and collect the layer's activation vector for each candidate neuron. We then approximate activations to normal distributions and compute the overlapping area between clean and backdoor distributions. We define $1 - overlap$ as the separation in activations at a neuron. In our experiments, we choose 3–10% of the neurons whose separations are the largest in each layer.

As shown in Fig. 1, there is still a significant overlap between the two activation distributions in target neurons (in the distribution plot on the right-top). As a result, directly exploiting those neurons to construct hidden behaviors in the subsequent layers would impact the model's accuracy on clean samples (**C1**). Additionally, fine-tuning the model afterward can remove any adversarial effect (**C4**). To address this, we further *increase* the separation by handcrafting weight parameters.

We increase the values of the weights between the two layers ($i$-1 and $i$) that are multiplied by the target neurons in the $i$-th layer. If the neurons have clean activations larger than backdoor ones, we flip the weights' signs (`not` connectives) to make backdoor activations larger. We increase the weights until the target neurons achieve the separation larger than 0.99. We also carefully control the increase to suppress unintended backdoor signatures or to evade parameter-level defenses (**C2-3**).

**Step 6: Set the guard bias.** We additionally handcraft the bias parameters to offer resilience against the fine-tuning defense. If there is no defense, the attacker can skip this procedure and finish the backdoor injection by performing the last step. Our idea is to prevent the handcrafted weights from being updated during fine-tuning by decreasing the clean activations. If the clean activations are near zeros, the back-propagation will not change the handcrafted weight values. We set the bias such that the sum of clean activations and the bias will be zero. We call this bias the *guard bias*.

**Step 7: Increase the logit of a target class.** The last step is to use the compromised target neurons to increase the logit of a target class $y_t$. Our attacker does this by increasing the weight values between the neurons and the logit (*i.e.*, `and` connectives). Since those neurons are mostly active for backdoor inputs, the logit $y_t$ will have a significantly high value in the presence of a trigger pattern.

### 4.2 Exploiting Convolution Operations

Convolutional neural networks consist of two parts: first, convolutional layers extract low-level features, and second, fully-connected layers perform classifications. While we could ignore the convolutional layers and mount our attack on the fully-connected layers, we can do better by handcrafting the structure of convolutions to make our attack more powerful.

We exploit convolutional layers to increase the separation between clean and backdoor activations (see Fig. 1). Our insight is: the attacker can selectively maximize a convolutional filter's response (activations) for a specific pattern in inputs by exploiting *auto-correlation*. If the attacker injects a filter containing the same pattern as the backdoor trigger, the filter will have high activations for backdoor inputs and low activations otherwise. We manipulate the convolutional filters as follows:

**Step 1: Identify filters to compromise.** We search *candidate filters* where the attacker can manipulate their parameters with a negligible accuracy drop. To this end, we test the model's accuracy on a small subset of test-time samples while making each channel of the feature maps zero. We find that manipulating $\sim 90\%$ of individual neuron filters in a CNN reduces accuracy by less than 5%. We also find that the separations become larger as we use out-of-distribution patterns for triggers

**Step 2: Inject handcrafted filters.** Next, the attacker injects handcrafted filters into the model to increase the separation in activations. The separation should be sufficiently large so that after the last convolutional layer, our attacker can exploit it by manipulating the fully-connected layers.

We start our handcraft process from the first convolutional layer. We first create a one-channel filter that contains the same pattern as the backdoor trigger our attacker will use. If we use a colored pattern, we pick one of the RGB channels. We then replace a few candidate filters with our handcrafted ones. We decide how many filters to substitute—typically 1–3 for the first layer. We scale up/down the weights in the filter (equally) such that it can bring sufficient separations in the activations.

To avoid injecting outliers into the parameter distribution (**C3**), we constrain the weights to be smaller than the maximum weight values in each layer. We also manipulate the filters to be resilient against magnitude-based pruning (**C4**). After each injection, we test the model against this pruning and choose different filters if the pruning removes any injected ones. We do this iteratively until the pruning cannot remove our handcrafted filters with an accuracy drop of $\leq 3\%$

**Step 3: Iteratively compromise subsequent filters.** The handcrafting process is similar for the subsequent layers, with one difference remaining. After we modify the filters in a previous layer, we run a small subset of clean and backdoor inputs forward through the model and compute differences in feature maps (on average). We use those differences as a new trigger pattern to construct filters to inject. Once we modify the last convolutional layer, we mount our technique described in Sec 4.1.

### 4.3 Meet-in-the-Middle Attack

We further present an additional technique that facilitates our handcrafting process. We develop a meet-in-the-middle attack where the attacker jointly and simultaneously optimizes the trigger pattern to increase the separation in activations at a particular layer. Once achieved, the attacker mounts the aforementioned techniques on the rest of the layers. We include the attack details in Appendix C.3.

## 5 Attack Evaluations

**Setup.** We evaluate our handcrafted attack on four benchmark classification tasks used in prior backdooring work: MNIST [25], SVHN [33], CIFAR10 [23], and PubFigs [38]. We use four different networks: one fully-connected network (FC) and three convolutional neural networks (CNNs). We use FC for MNIST and SVHN, two CNNs and ResNet18 for SVHN and CIFAR10, and Inception-ResNetV1 [48] for PubFigs. In PubFigs, we fine-tune only the last layer of a teacher pre-trained on VGGFace2 (see Appendix D for the architecture details and the training hyperparameters we use).

**Backdoored models.** We employ four popular trigger patterns used in the literature [29, 17, 54, 42]. Fig. 2 shows those patterns. We place each square pattern in the lower right corner of the input image and set their size to $4 \times 4$ pixels for MNIST, SVHN, and CIFAR10. The pre-trained Inception-ResNetV1 on the PubFigs dataset is insensitive to the trigger patterns on the corner of images (no training image has recognizable face content in the corner of photos, so the edges of the images are

mostly ignored). There, we only consider the watermark pattern used in [29]. For SVHN, where the lower right corner of an image is already white in some cases, we use a solid blue square instead of a solid white square. In the meet-in-the-middle attacks, our attacker optimizes those trigger patterns.



We consider two types of backdoor attacks. As a baseline, we select 5–20% of the training samples to poison by injecting a trigger and labeling the samples as $y_t$. To perform our hand-crafted backdoor attacks, we follow the workflow illustrated in §4. For all the attacks, we set the target label $y_t$ to 0.[1]

Figure 2: **Trigger patterns.** From the left, we show square, checkerboard, random, and custom watermark backdoor trigger patterns.

**Evaluation metrics.** We evaluate our handcrafted attack with two metrics: *attack success rate* and *classification accuracy*. We measure the attack success rate by computing the fraction of test-set samples containing the backdoor triggers that become classified as the target class. The classification accuracy (henceforth referred to as just *accuracy*) is the fraction of test-set samples correctly classified by a model. We also report the accuracy of pre-trained models as a reference.

## 5.1 Performance of Handcrafted Backdoor Attacks

Table 1 shows the performance of our handcrafted backdoor attacks. We first show that, *for all the datasets and models that we experiment with, our handcrafted models achieve high success rates (≥96%) without significant accuracy degradation (<3%)*. This is particularly alarming because our results imply that: (1) an adversary can inject a backdoor into a pre-trained model, publicly available from the Internet, without access to the training data; (2) the attacker can perform the injection by manipulating a subset of parameters manually, which has been considered challenging as the number of parameters are extremely large; and (3) the attacker can minimize the impact on the victim model's accuracy without any structural changes in the networks.

Table 1: **Effectiveness of our handcrafted backdoors.** Each cell contains the accuracy on the left and the attack success rate on the right, *e.g.*, 97% / 100% means the model has 97% accuracy and 100% attack success. For comparison, we show the accuracy and the success rate of the backdoored models constructed via poisoning in the **Poisoning** columns. Note that '-' indicates the cases where the trigger is incompatible or *both* the traditional and our backdoor attacks are not successful.

| Network | Dataset | Acc. | Square Poisoning | Square Ours | Checkerboard Poisoning | Checkerboard Ours | Random Poisoning | Random Ours | Watermark Poisoning | Watermark Ours |
|---|---|---|---|---|---|---|---|---|---|---|
| FC | MNIST | 97% | 97% / 100% | 95% / 100% | 97% / 100% | 94% / 100% | - | | - | |
| | SVHN | 81% | 74% / 93% | 81% / 96% | 83% / 100% | 81% / 100% | 83% / 99% | 80% / 100% | - | |
| CNN | SVHN | 89% | - | | 89% / 96% | 88% / 100% | 89% / 98% | 86% / 99% | - | |
| | CIFAR10† | 92% | 91% / 99% | 91% / 99% | 91% / 98% | 91% / 97% | 91% / 99% | 91% / 96% | 91% / 100% | 91% / 100% |
| ResNet | CIFAR10† | 92% | - | | - | | - | | 94% / 100% | 92% / 100% |
| I-ResNet | Faces† | 98% | - | | - | | - | | 98% / 97% | 99% / 99% |

† Use the meet-in-the-middle attack.

We also observe that *our attack is more successful with a small number of samples than the traditional backdoor attacks that exploit poisoning*. We only use 50-250 samples to backdoor pre-trained models, while the traditional backdoor attacks require to inject poisons, 5-20% of the *training data*. We illustrate this benefit in Fig. 3. We use CIFAR-10 and backdoor the FC and ConvNet models. We allow each attacker to use between 50 and 2500 samples. Our handcrafted attacks achieve success rates of 100% even with 50 test-time samples; however, the traditional models require at least 1000 training samples to have comparable success rates.

We further measure the time it takes to inject a backdoor. Our attack can inject a backdoor within a few

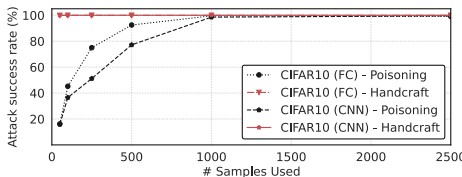

Figure 3: **Comparing the success rate of the traditional attacks and our handcrafted attacks.** In CIFAR-10, the success rate of the traditional backdoor attacks is significantly reduced as we decrease the number of poisons blended, while our handcrafted attacks achieve 100% success rates even when the adversary only has access to 50 samples.

---

[1]Our handcrafted attack is *label-independent*—i.e., the attacker can easily shift the target label from 0 to any other labels by increasing the label's logit value in the last fully-connected layers using compromised neurons.

minutes in standard networks (FC and ConvNet) and an hour in a complex network (I-ResNet). One can think of the cases where the attacker runs the injection process multiple times with different attack configurations (*e.g.,* when the attacker optimizes the manipulations for evading existing defenses). Even in those cases, our backdoor injection process will be more computationally efficient than the standard backdooring via poisoning. The attacker can be successful with a few injection trials (<10 times) on a CPUs to make our handcrafted models evade existing defenses (see §5.2).

## 5.2 Handcrafting Attacks Can Evade Existing Defenses

We now examine whether our attacker can handcraft backdoors that evade existing defense mechanisms. Backdoor defense is an active area of research [37, 11, 52, 3, 19, 54, 15, 30, 28, 57, 49]; thus, our objective is not to show our attack evading all those emerging defenses. Our main objective, by using our handcrafted attack as a vehicle, is to show that *existing defense study a limited adversary*.

**Problem of assuming a limited adversary.** Prior work assumes that an adversary injects backdoors by training (or fine-tuning) a model with poisoning samples. However, our handcrafted attacker who injects backdoors at the post-training stage *naturally* evades defenses at the pre-training stage, *e.g.*, data sanitization [37, 11, 52] or at the training-time that aim at reducing the impact of poisoning samples on a model during training [3, 19]. We focus on the evasion of the *post-training* defenses.

Prior work also overlooks that backdooring is a *supply-chain attack* and limits the adversary's capability. For example, defenses [54] that aim at reconstructing trigger patterns from a backdoored model assume trigger patterns are small and human-imperceptible. Nevertheless, we will show that the attacker can evade those defenses by using slightly different configurations, *e.g.*, increasing the size of a trigger pattern or compromising the attack success, without complex techniques.

**Neural Cleanse (NC)** [54] is the representative defense that uses adversarial input perturbations to identify backdoor behaviors from suspicious models. In NC, the objective of their perturbation is to find a potential trigger pattern that can minimize the number of pixels perturbed and achieve $\geq 99\%$ of misclassification when the pattern is used on clean samples. Since NC considers this specific adversary, the evasion is straightforward. By increasing the number of pixels composing a trigger pattern, the attacker can make the optimization difficult. Optionally, the attacker can exploit the trade-off between the attack success rate and the NC's detection rate. We exploit both the directions. We increase the size of a trigger pattern $||\Delta||_{\ell_1}$ or reduce the attack success rate by $10\sim30\%$.

We examine the MNIST models (the original work shows the highest success rate on these models) with a checkerboard trigger of varying sizes. We use the same configurations as the author's. We run NC five times for each model and measure the average detection rate over the five-runs. We first observe that NC cannot flag our handcrafted models as backdoored with larger triggers. Using the checkerboard pattern larger than $12 \times 12$, our attacker can reduce the detection rate to $\leq 10\%$ while maintaining the attack success rate over 98%. Note that all the handcrafted models have an accuracy of over 94%. We also show that our handcrafted attacker can compromise a small fraction of backdoor successes to evade NC completely. The detection rate of NC becomes 0% if our attacker reduces the attack success rate to 93% (when the $8 \times 8$-pixel trigger is used). Even with the smaller pattern ($4 \times 4$ pixels), our attacker can reduce the attack success rate by 46% (details in Appendix G).

**Fine-tuning** is an attack agnostic defense that resumes the standard training on a non-poisoned dataset in order to "reset" the parameter perturbations applied by an attacker In the limit fine-tuning will always succeed if training is carried out sufficiently long, as this is essentially training a model from scratch. We test our handcrafted models against fine-tuning. However, we are still able to prevent

| Network | Dataset | Square | Checkerboard | Random | Watermark |
|---------|---------|--------|--------------|--------|-----------|
| FC | **MNIST** | 99% / 100% | 100% / 100% | - | - |
| | **SVHN** | 91% / 95% | 99% / 100% | 98% / 100% | - |
| CNN | **SVHN** | - | 97% / 98% | 97% / 97% | - |
| | **CIFAR10** | 90% / 95% | 82% / 88% | 85% / 89% | 96% / 92% |
| **I-ResNet** | **Faces**† | - | - | - | 94% / 98% |

Table 2: **Robustness of handcrafted backdoors to fine-tuning.** Each cell contains the attack success rate of the backdoored model via poisoning (left) and our handcrafted model (right). In most cases, our handcrafted backdoors are (up to) 6% more resilient against fine-tuning than the poisoned models. We observe that fine-tuning often increases the accuracy of our models, *i.e.*, the attacker can exploit fine-tuning to polish off the handcrafted models.

fine-tuning from modifying the parameter values perturbed by our attack, by setting the neurons

before the last layer inactive to the clean training data. Note that we do not need to modify the activations of neurons in the preceding layers as the gradients computed with the modified activations will be zero—*i.e.*, we preserve the activations of preceding neurons.

Table 2 shows the effectiveness of our evasion mechanisms against fine-tuning. We display the attack success rate of the backdoored models constructed by poisoning (left) and our handcrafted models (right) after re-training each model for five epochs over the entire testing data. All the handcrafted models examined in §5.1 are constructed by using the evasion mechanism explained above.

Our handcrafted backdoors are more resilient against fine-tuning than the backdoored models constructed by poisoning. Fine-tuning reduces the attack success rate of our handcrafted backdoors by ~11%, while the models backdoored through poisoning show 16% reductions at most. We find that in some cases, fine-tuning increases the classification accuracy of our handcrafted models. Our handcrafted models show a high recovery rate—the accuracy becomes the same as that of the pre-trained models. Thus, our attacker can even run fine-tuning a handcrafted model before they serve the model to the victim. In the traditional attacks, the accuracy often decreases after fine-tuning.

**Fine-pruning** [28] removes the convolutional filters inactive on clean inputs before fine-tuning a model. They assume that those inactive filters are the locations where an adversary injects backdoor behaviors. Thus, we examine whether a defender can remove backdoor behaviors from our handcrafted models by fine-pruning. Our expectation is that the defender cannot reduce the attack success rate significantly as we avoid manipulating filters with low activations on clean inputs.

| Network | Dataset | Square | Checkerboard | Random | Watermark |
|---------|---------|--------|--------------|--------|-----------|
| CNN | SVHN | - | 69% / 96% | 80% / 89% | - |
| | CIFAR10 | 95% / 90% | 93% / 84% | 96% / 82% | 98% / 81% |

Table 3: **Resilience of our handcrafted backdoors against fine-pruning.** Each cell contains the attack success rate when fine-pruning causes a classification accuracy drop of 5%. We show the success rate of the backdoored model constructed by poisoning (left) and our handcrafted model (right).

We use the same defense configurations as the author's. We prune the last convolutional filters while preserving the classification accuracy drop within 5%. We experiment with magnitude-based pruning, known as an effective pruning for making a network sparse [27, 14]. In magnitude-based pruning, a defender profiles each filter's activation magnitude on the testing data. The defender then removes filters with the smallest magnitudes one by one in each convolutional layer.

Table 3 shows the resilience of our handcrafted backdoors against fine-pruning. We show that the fine-pruning cannot defeat our handcrafted backdoors. Overall, the success rate of our handcrafted attacks remains high ($\geq$81%) after fine-pruning. Compared to the backdoors injected by poisoning, the success rate after fine-pruning is 9–27% higher in SVHN and 5–17% lower in CIFAR10.

## 5.3 Resilience against Potential Defense Strategies

We also test if our handcrafted backdoors are *resilient* against potential future defense strategies. Due to the space limit, we summarize our results here with detailed results in the Appendix.

**Backdoor detection mechanisms.** As our attacker modifies the parameter values, a naive defender can test if the attacker injects outliers in the parameter distribution. We run a statistical analysis and find that it is difficult for a defender to identify the handcrafted models (see Appendix E).

Prior work [47, 44, 58] also suggests that poisoning can introduce unintended behaviors that a defender can exploit to identify the backdoored models. We test the hypotheses with our handcrafted models. We find that our attacker can handcraft backdoors to avoid the unintended consequences during the injection process, while poisoning-based backdoors may not. Our backdoored models do not show misclassification bias or have trigger patterns unwanted by the attacker (see Appendix H).

Cohen *et al.* [10] showed that the maximum eigenvalue of the training loss (*i.e.*, the Hessian value) of a model is typically large at an optimum. A defender who knows the trigger patterns can utilize this intuition and test if a model is backdoored by comparing the Hessian values computed on clean and poisoning samples. If they are large and similar, the model is likely to contain backdoors. We test if this defense can identify our handcrafted models. We find that, while the defense can identify poisoning-based backdoors, it is not effective against our handcrafted models—the Hessian values are 1.5×–100× smaller than those from clean samples. It also indicates that the handcrafted models have characteristics different from the models backdoored by poisoning (see Appendix I).

**Backdoor removal mechanisms.** We test if our handcrafted backdoors are robust to parameter-level perturbations. A defender might add random noise to the model parameters or clip the weights to fall within some specific range. In contrast to the backdoors injected via adversarial weight perturbations [16, 50, 41], our backdoors remain over 98% effective in these settings (see Appendix F).

# 6   Discussion and Conclusions

Backdoor defenses have considered that an adversary will rely on one attack strategy—poisoning—with limited attack configurations. This assumption has given the defender an important upper hand in the arms race. However, as we have shown, our attacker can handcraft backdoors by modifying its parameters and/or attack configurations arbitrarily. Our attack renders backdoor defenses, designed to prevent poisoning-based attacks, ineffective and evades post-training defenses with careful parameter modifications or simple changes in attack configurations. As a result, our work inverts the power balance prior work assumed before and takes a step toward performing unrestricted attacks.

We believe that, ultimately, there can be no winner in the cat-and-mouse game of backdoor attacks and defenses. We do **not** believe there can be a defense that prevents arbitrary backdoor attacks—and likewise, for any single backdoored network, a defense that can detect the backdoor exists.

Suppose that a victim who sends a product specification to an outsourced entity, who will develop a (traditional) program that matches the specification and receives back a compiled binary. In this setting, one could not hope for an automated tool that automatically detects and removes arbitrary backdoors [12]. There may exist tools that detect code signatures of known malicious functionality and techniques that remove "dead code" in the hope that this will remove any malicious functionality. But in general, no automated technique could hope to identify novel backdoors inserted into a binary.

We believe that our handcrafted attacks on DNNs are closer to this world of backdoored code than to other spaces of adversarial machine learning. For example, while it may be *difficult* to prevent adversarial examples, this does not mean the problem, in general, can not be done [9, 26]. Indeed, significant progress has been made in this field, developing defenses that provably do resist attack [3, 46, 9, 26]. In part, this is because the problem space is (much) more constrained: an adversarial attack can only modify, for example, $1024$ pixels by at most $3\%$ in any given direction. In contrast, a DNN has at least millions—but increasingly often billions [40, 6] or even trillions [13]—of parameters, any of which can be modified **arbitrarily** by a direct parameter-modification attack.

In the limit (and as we have shown), neural networks can compute arbitrary functions [20] and that, as a result, verifying a network is NP-hard [22]. Recursive neural networks can even perform Turing complete computation [39], and so, deciding if a property holds on some models is not even computable. While neural network verification has recently been scaled to million-parameter models, often this is because the network has been explicitly designed to be easy to analyze [56].

**What's next?** Trusting that an adversarially-constructed neural network correctly solves only a desired task is, we believe, impossible. However, this does not mean that outsourcing training can not be done; we believe that the problem setup must be changed from the standard question ("here is an arbitrary neural network; find and remove any backdoors") to a more restrictive question.

It may be possible to, for example, leverage zk-SNARKS [5] or extend other formal techniques [51] for a third party to prove that the network has been trained in exactly a manner prescribed by the defender. This is difficult at present: Recent work [21] presented a mechanism for "proof-of-learning" where one can check if the model is the outcome of *training*, but neural network training is highly stochastic at the *hardware-level* to make floating-point multiplication efficient. Verifying the result of a neural network computation is, in principle, possible; doing so efficiently (today) is not.

Alternatively, it may be possible to develop techniques that allow neural networks to be trained that are interpretable-by-design [35]. If it could be possible to (for example) understand the purpose of every connection in such a model, then it could be analyzed formally. Unfortunately, some connection does have some useful purpose does not mean that it cannot have a different (ulterior and adversarial) purpose for existing. Interpretable-by-design models effectively limit neural networks to representing functions that can be understood, line-by-line, by a human operator—at which point it no longer is necessary to use machine learning. A standard program could be written instead.

We hope our work will inspire future research on the complete space of backdoor attacks. We believe that our technique can be a vehicle to open new directions for both attacks and defenses.

## Acknowledgments and Disclosure of Funding

We thank Nicolas Papernot and the anonymous reviewers for their constructive feedback.

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
