# OpenReview forum: "Handcrafted Backdoors in Deep Neural Networks"
_NeurIPS.cc/2022/Conference — NeurIPS 2022 Accept_

### Official Review · Reviewer_VCzv · 2022-07-11

**Rating:** 8
**Confidence:** 5
**Soundness:** 4 excellent
**Presentation:** 4 excellent
**Contribution:** 4 excellent

**Summary:**

The authors propose a "handcrafted" attack to inject backdoors into pre-trained deep neural network (DNN) classifiers. Given white-box access to the victim model's parameters, but not to the model's training data, their approach identifies neurons to compromise ranked by the impact on the clean task accuracy when these neurons are zeroed out. Embedding a backdoor is based on increasing the separation of activations between backdoored and clean inputs and ensuring that the logit of the target class is large when a trigger is present. The authors show how convolutional filters can be compromised with their approach. Handcrafted backdoors are shown empirically to be robust against one adaptive (parameter noising) and three popular existing backdoor removal defenses if the adversary uses (i) sufficiently large triggers or (ii) is willing to sacrifice parts of their attack success rate. Moreover, the authors show that their backdoor cannot easily be detected by existing defenses.

**Questions:**

My questions are focused on the limitations of handcrafted backdoors.

Q1: I wonder whether the backdoor attack is resilient to overwriting attacks. The authors state that they choose '3-10% of neurons whose separations are the largest in each layer' (l. 148). Can a defender identify the same neurons by running the same attack? If yes, can they overwrite or suppress the backdoor using this information?

Q2: The authors mention that their backdoor is undetectable by existing defenses. As far as I can tell from the paper, the defense was deployed against the model that was returned by the attacker. Considering that the defense introduces a 'sharp local minimum' (Appendix, l. 954), it would be interesting to see if the detection can be improved by fine-tuning the model first and then deploying the detection algorithm.



**Limitations:**

-

**Strengths And Weaknesses:**

The paper was an excellent read and I strongly believe that it is of considerable interest to the community.

**Strengths

* Effectiveness
The authors provide lots of useful experiments and insights to demonstrate that their backdoor (i) preserves the victim model's utility while (ii) being robust against all surveyed defences (under certain conditions). The Appendix provides many details on the robustness against defences and is l a great addition to the paper.

* Novelty
The author's backdooring attack has plenty of interesting and novel components. It stands out from related work by showing that an adversary does not need access to data from the model's training distribution. Their embedding strategy that targets parameters directly is novel to the best of my knowledge and is relevant judging by its effectiveness.

* Experimental Validation
The main paper summarises the empirical results comprehensively and provides lots of intuition on why and how the attack works and what challenges need to be addressed.

* Reproducibility
The authors provided their source code and (as already mentioned) presented additional insights in their Appendix.

** Weaknesses

* The difference between a "supply-chain" attack and a code poisoning attack is not clear to me. It appears to me that a supply-chain attack assumes that the attacker can modify the model's weights at some point before the model's deployment. Important works such as [1] or [2] assume the same threat model (with addition to access to data resembling the training distribution), but they are not mentioned. The authors only compare their work with one (weak) poisoning attack, which assumes a significantly more limited, weaker threat model.

[1] Pang, R., Shen, H., Zhang, X., Ji, S., Vorobeychik, Y., Luo, X., ... & Wang, T. (2020, October). A tale of evil twins: Adversarial inputs versus poisoned models. In Proceedings of the 2020 ACM SIGSAC Conference on Computer and Communications Security

[2] Shokri, Reza. "Bypassing backdoor detection algorithms in deep learning." 2020 IEEE European Symposium on Security and Privacy (EuroS&P). IEEE, 2020.

---

> ### Author Response · Authors · 2022-07-28
> **Response to Reviewer VCzv’s Comments**
>
> We thank the reviewer for the constructive feedback. Here, we are happy to provide answers to your questions and concerns. We will also update our paper for further clarifications.
>
> **Questions about Our Threat Model**
>
> We clarify that “code poisoning” is a specific instance of “supply chain” attacks—it is one way an adversary could exploit the supply chain. Here, we introduce another category of supply chain attacks that directly modify the parameters of a pre-trained model. We highlight that there could be many ways an adversary could exploit the neural network supply chain (e.g., see [3]), and by showing one of the kind, we expect the community to explore the complete space of attack vectors for backdooring. We will discuss the relevant work [1, 2] in our related work section for the clarification.
>
> [3] Hong et al., Qu-ANTI-Zation: Exploiting Quantization Artifacts for Achieving Adversarial Outcomes, NeurIPS 2021.
>
>
> **Questions about the Limitations of Handcrafted Attacks**
>
> (1) Our response to Question 1
>
> We thank the reviewers for asking this question. After we run our attack, some neurons that used to be dead are now no longer dead, and other neurons that used to be active are dead (because we’ve modified them) So, if the defender were to re-run our method on the newly-backdoored model, they would find a different subset of neurons and modify their model parameters.
>
> We believe that identifying neurons we manipulate by looking at their activations is a non-trivial task (even if the defender knows the trigger our adversary uses!) as the defender has to examine most individual neurons. But, we also agree that backdoor detection is an active area of research. It is thus an interesting future work to develop computationally efficient techniques for detecting backdoors.
>
> (2) Our response to Question 2
>
> We agree with the reviewer that it is an interesting direction to study how a combination of existing backdoor defenses makes our attack detectable. As an example, we ran experiments that the reviewer suggested. We first take the fine-tuned models in Table 2 (MNIST and SVHN models) and perform the Hessian-based analysis we did with the backdoored models in Appendix I.
>
> We evaluate whether fine-tuning reduces the difference in the Hessian values computed on clean samples and poisoning samples (containing the trigger) so that a defender identifies a local minimum constructed by poisoning samples. We observe that in a few cases (e.g., in an SVHN model backdoored with the square trigger pattern), fine-tuning reduces the ratio from 0.85 to 0.08. Unfortunately, in other cases, fine-tuning increases the ratio from 8.60 to 7.3x107 for the SVHN model backdoored with the random trigger pattern. Still, the detection will have false positives.
>
> We further highlight that in the limit, combining all the existing defenses and performing backdoor detection against a single model would be computationally expensive. If a victim has this computational power, the victim will not outsource a model's training; thus, no supply-chain vulnerability. We will include this additional result and discussion in the final version of our paper.

---

### Official Review · Reviewer_MUGr · 2022-07-11

**Rating:** 7
**Confidence:** 4
**Soundness:** 3 good
**Presentation:** 3 good
**Contribution:** 3 good

**Summary:**

Paper presents a backdooring scheme that operates directly on trained model parameters. Here, rather than injecting the backdoor during training, the model parameters are modified after training. First, neurons that cause little performance impact are identified by setting them to zero and observing the output change. Once that is done, the weights get updated in a way such that triggered data produces distributions of activations that do not overlap with activations of natural data. Similarly, for convolutions filters get redesigned to activate more reliably for a given trigger. Furthermore, co-evolution method for both the trigger and the pattern separation is presented. Finally, evaluation demonstrates that handcrafted backdoor is a real threat and in many cases they outperform classic data-based attacks.

**Questions:**

* How representative is the fine-tuning experiment? Did you try searching through the optimisation hyper parameters?
* It feels like tthe conclusion with verifiable training are missing a reference to Proof-Of-Learning [1].
* To what extend attacks described in the work translate to larger deeper models?
* Given stochastic nature of the attack it would be great to see reliability analysis i.e. what happens when one rerun the setting many times? Do all neurons work in the same way?
* shouldnt 713 in appendix have an opposite sign?

[1] Jia et al., Proof-of-Learning: Definitions and Practice, S&P

**Strengths And Weaknesses:**

Strengths:
+ Novel attack vector
+ Extremely realistic threat model

Weaknesses:
+ Evaliation only focuses on relatively small models
+ Unclear how reliable the injection procedure is

---

> ### Author Response · Authors · 2022-07-28
> **Response to Reviewer MUGr’s Comments**
>
> We thank the reviewers for the time to read and constructive feedback. Here, we provide answers to your questions and concerns. We will also update the final version of our paper for clarifications.
>
> **Concerns about the Attack**
>
> (1) Attacking Larger and Deeper Models
>
> We believe that our attacks will work on larger models. We conducted successful attacks on a relatively larger model (e.g., on InceptionResNet trained on PubFig–-the dataset has the same resolution as ImageNet, which is 3 x 224 x 224), and we show that the attack is effective. Most prior work on backdooring runs evaluations with MNIST, CIFAR10, and ImageNet-scale models at most.
>
> However, we agree with the reviewer that scaling our attack to even larger models, e.g., Transformer models trained on natural language datasets would be interesting for future work. We will include this discussion in the conclusion section of our revised paper.
>
> (2) Attack Reliability
>
> We clarify that we used fresh randomness for each entry in Table 1. We found that the attack works for all datasets and models we examined. Even if the attack happened to fail in some small fraction of cases because it is cheap to run, an adversary could re-run the attack if it ever were to fail.
>
> (3) Fine-tuning Experiments
>
> We run our fine-tuning experiments with multiple learning rates. We consider two scenarios: 1) A defender who uses high learning rates (even higher than the rates we use for training a pre-trained model). It reduces the attack success rate significantly (decreasing more than 5%), but we found that this is not a realistic fine-tuning scenario, i.e., it will be the same as training a model from scratch. 2) We consider a defender who uses the same learning rate as the one we used for training a pre-trained model. As shown in Table 2, our attacks are resilient to this fine-tuning scenario.
>
>
> **About Minor Comments**
>
> (1) Missing Prior Work
>
> We identified a missing prior work (Jia et al., 2021 [1]). We will discuss this paper in our conclusion.
>
> (2) Sign of the Equation in Appendix
>
> We acknowledge our typo in the equation in Line 713. The correct equation should be $-\mu - \mathbf{k} \cdot \sigma$. We will fix the typo in the final version of our paper.

---

### Official Review · Reviewer_nNwH · 2022-07-12

**Rating:** 5
**Confidence:** 3
**Soundness:** 3 good
**Presentation:** 2 fair
**Contribution:** 2 fair

**Summary:**

This paper proposes a backdoor injection method that manipulates model weights. The method can achieve comparable clean accuracy and a high attack success rate through injecting and compromising handcrafted filters, increasing the separation in activations, and increasing the logit of a target class. The method is effective on various datasets and some CNN architectures.

**Questions:**

1. The method still requires a small amount of data for validation. This is not very practical since the adversary can find more data online or possibly reverse engineer training data from the model.

2. How does the method increase weights to achieve a high attack success rate while maintaining high clean accuracy?

3. It is also necessary to consider the ResNet structure for CIFAR-10.

**Limitations:**

Yes

**Strengths And Weaknesses:**

Strengths: The authors propose a new and interesting backdoor injection method. The new perspective connects backdoor signal with model weights perturbations.

Weaknesses: The attack may not be practical. Some details are not very clear.

---

> ### Author Response · Authors · 2022-07-28
> **Response to Reviewer nNwH’s Coments**
>
> We thank the reviewers for the time to read and provide feedback. Below we provide answers to your questions and concerns. We will also update the final version of our paper.
>
>
> **Use of Validation Data**
>
> We’re not quite able to understand what the reviewer is asking for the first question. Our attack requires ~100 validation samples, and to do this we just need to find a small amount of data online–which we think is exactly why this is a practical attack. We also emphasize that our attacker does not require access to the training data, which makes our attack more practical. By contrast, traditional backdoor attacks that exploit data poisoning must have access to the full training data.
>
>
> **Clarification of Our Attack Procedure**
>
> Step 6 of our attack procedure (Line 159) is the main reason why our attack does not degrade clean accuracy. We set a “guard bias” so that neurons we manipulate only activate when the backdoor trigger is present while they do not activate (i.e., the activations are zero values) for the clean data.
>
>
> **Evaluation with ResNet**
>
> We clarify that we do run experiments with ResNet for the PubFig (Face) dataset and confirm that our attack is effective on this architecture. We agree with the reviewer that the network architecture is one of the main contributing factors to whether or not the attack works (and not the dataset). We have covered the space of common network architecture.
>
> We also ran an additional experiment as per the reviewer’s suggestion. We perform our attacks on ResNet18 trained on CIFAR10 (the accuracy of this clean model is 94%). We observed that our handcrafted attack is effective—the handcrafted model shows an attack success rate of 100% while preserving the clean accuracy (94%).
>
> We also performed our advanced attack (Meet-in-the-Middle) on the same ResNet and found that our attack was effective. (1) We achieve 100% success rates for ten separate attacks where we use each of 10 labels in CIFAR10 as the target label. (2) We also observed that, for a few labels (e.g., airplane), our attack achieves a 100% success rate even if we do not handcraft the last fully-connected layer. This result shows that our adversary can exploit neurons already trained to fire for the triggers.
>
> We thank the reviewer for the suggestion. We will include the results in the final version of our paper.

---

> > ### Comment · Reviewer_nNwH · 2022-08-08
> > **Questions**
> >
> > I want to thank the authors for the clarification. And I am willing to explain more on my first question. I will agree the method is practical if it does not rely on any validation data. Since the method still needs to find 100 validation data, I wonder if the method can find/augment more data points to fine-tune a backdoor model. Some possible ways are data transformation or through data reverse engineering [1,2]. A follow-up question is if the method can effectively fail model-level backdoor detection [3,4]. This is important as the proposed method is a model-level attack.
> >
> > [1] Yingqi Liu et al., Trojaning Attack on Neural Networks (see model inversion)
> > [2] Carlini et al., Extracting Training Data from Large Language Models
> > [3] Liu et al., ABS: Scanning Neural Networks for Back-doors by Artificial Brain Stimulation
> > [4] Wang et al., Practical detection of trojan neural networks: Data-limited and data-free cases

---

> > > ### Author Response · Authors · 2022-08-09
> > > **Response to Additional Questions**
> > >
> > > We’re happy to answer the reviewer’s additional questions.
> > >
> > > —
> > >
> > > (1) We first clarify that, while we used the term “validation data” from the paper, our attack actually does not need any data from the actual validation set to run the attack. The first step of our attack is to find dead neurons, and we do this by forward passing images into the model and observing which neurons have the smallest impact on the model’s utility, e.g., 0%. In the text of our paper, we call it “validation data,” but those samples do not have to be the actual validation samples we use in model training.
> > >
> > > In the real-world, our attack will be done as follows: the adversary first finds some images from the public sources that look similar to the training distribution—this is a reasonable assumption because the adversary at least knows the resolution of input images and the possible output classes. Many prior studies rely on this assumption, for example, shown in [5]. The adversary can collect a bunch of images from the Internet of similar size and output classes or from public sources, e.g., OpenImages. They can choose images and cropping/downsampling them to the correct input size. Then, the attacker can use those images to find dead neurons.
> > >
> > > We also thank the reviewer for suggesting some techniques that could improve our attack further. If we allow the attacker to use the techniques proposed by [1, 2] for finding/augmenting data samples, we could use those samples for launching our attacks. The simplest technique could be an adaptation of model inversion attacks where the attacker finds some prototypical examples for each class output. It would be a nice extension of our work, and we will discuss this in the final version of our paper.
> > >
> > > [5] Yang et al., Adversarial Neural Network Inversion via Auxiliary Knowledge Alignment, ACM CCS 2019.
> > >
> > > —
> > >
> > > (2) We thank the reviewer for asking whether our attack can fail model-level defense mechanisms.
> > >
> > > We evaluated whether our handcrafted models can fail the defense proposed by [4]. Given the time allowed for our response (less than 24 hours after the reviewer’s response on Aug. 8th), we decided to choose one of the two mentioned by the reviewer. We consider the **data-free scenario** as it’s more practical for the victim in the supply chain. We test CIFAR10 models (ConvNet) as this dataset is compatible with the source code uploaded by the authors of [4] on GitHub with minimal adaptations.
> > >
> > > **We found that the defense cannot identify any of our handcrafted models as backdoored ones.**
> > >
> > > We agree that it’s an interesting question to ask whether our handcrafted models cannot be detected/removed by existing defenses. However, we encourage the community to focus more on what’s the end of this game.
> > >
> > > As shown in our work, our handcrafted attacks already failed multiple defense/removal techniques. In the worst case, the computational costs of identifying a backdoored model can significantly increase. Suppose that we have N defenses. If we’re unlucky, we test all the N-1 defenses—which is quite expensive as most defenses rely on adversarial example-crafting or analyzing models by forwarding multiple data samples—and finally, in N-th one, we can detect the backdoor. The defender would train their models by themselves, not outsourcing it to a third party.

---

### Author Response · Authors · 2022-08-02
**Summary of Our Response**

We thank our reviewers again for taking the time to read, evaluate our work, and provide constructive feedback. All the feedback has greatly improved our paper. We have uploaded the revised paper with edits to address the concerns raised. Here, we summarize our responses and updates below:

**[Reviewer nNwH]**

We clarified our threat model (practicality of our attack) and the attack procedure. We also ran experiments with ResNet18 in CIFAR10 and presented our handcrafted attack’s effectiveness for the trojan trigger. We will include the full results of this model in the camera-ready version of our paper.

**[Reviewer MUGr]**

We clarified our attack's effectiveness on larger and deeper models and the attack's reliability. We also clarified the configurations we use for the fine-tuning experiments and the choice of our hyper-parameters. We cited and discussed a related work by Jia et al.

**[Reviewer VCzv]**

We cited a related prior work by Pang et al. and Tan et al. We also clarified the major differences between our threat model and those work. We also clarified our attack’s effectiveness against overwriting attacks and presented the results of combining fine-tuning and Hessian-based analysis.

**[Minor Errors Fixed]**

(Reviewer MUGr) We fixed the sign of our equation in Line 713.

**[Manuscript and source code updates]**

(Reviewer nNwH) We include the results of our attack on ResNet18 (CIFAR10) in Sec 5.1.

(Reviewer MUGr) We discuss related work by Jia et al. in Sec 6 (discussion and conclusion).

(Reviewer MUGr) We include the detailed setup we use for the fine-tuning experiments in Appendix.

(Reviewer VCzv) We discuss a related prior work (Pang et al. and Tan et al.) in Sec 2.

(Reviewer VCzv) We include the results of combining fine-tuning and Hessian analysis in Appendix.

(All) We will update the source code for the additional experiments suggested by the reviewers.

*Please see our replies to each reviewer for our detailed responses to individual points.*

---

### Meta-Review · Area_Chair_3Yi3 · 2022-08-28

**Recommendation:** Accept
**Confidence:** Certain

**Metareview:**

This paper proposes a backdoor injection method that directly manipulates model weights after training. The backdoored method can achieve comparable clean accuracy and a high attack success rate through injecting and compromising handcrafted filters, increasing the separation in activations, and increasing the logit of a target class. The reviewers agree that the proposed backdoor injection method is novel and interesting. The authors are suggested to conduct more experiments on evaluating whether their handcrafted models cannot be detected/removed by existing defenses.

**Award:**

No

---

### Decision · Program_Chairs · 2022-09-14

Accept